# The Anemia Stress Index—Anemia, Transfusions, and Mortality in Patients with Continuous Flow Ventricular Assist Devices

**DOI:** 10.3390/jcm11154517

**Published:** 2022-08-03

**Authors:** Supriya Shore, Thomas C. Hanff, Jeremy A. Mazurek, Arieh Fox, Monique S. Tanna, Edward W. Grandin, Robert Zhang, Joyce Wald, Carli Peters, Michael A. Acker, Pavan Atluri, J. Eduardo Rame, Lee R. Goldberg, Mariell Jessup, Kenneth B. Margulies, Edo Y. Birati

**Affiliations:** 1Division of Cardiovascular Medicine, University of Michigan, Ann Arbor, MI 48109, USA; shores@med.umich.edu; 2Division of Cardiovascular Medicine, University of Pennsylvania, Philadelphia, PA 19104, USA; thomas.c.hanff@gmail.com (T.C.H.); jeremy.mazurek@pennmedicine.upenn.edu (J.A.M.); monique.tanna@pennmedicine.upenn.edu (M.S.T.); joyce.wald@pennmedicine.upenn.edu (J.W.); carli.peters@pennmedicine.upenn.edu (C.P.); lee.goldberg@pennmedicine.upenn.edu (L.R.G.); kenneth.margulies@pennmedicine.upenn.edu (K.B.M.); 3The Zena and Michael A. Wiener Cardiovascular Institute, Icahn School of Medicine at Mount Sinai, New York, NY 10029, USA; arieh.fox@mountsinai.org; 4Division of Cardiology, Beth Israel Deaconess Medical Center, Boston, MA 02215, USA; wgrandin@bidmc.harvard.edu; 5Department of Internal Medicine, University of Pennsylvania, Philadelphia, PA 19104, USA; robert.zhang@nyulangone.org; 6Division of Cardiovascular Surgery, University of Pennsylvania, Philadelphia, PA 19104, USA; michael.acker@pennmedicine.upenn.edu (M.A.A.); pavan.atluri@pennmedicine.upenn.edu (P.A.); 7Jefferson Heart Institute, Thomas Jefferson University Hospital, Philadelphia, PA 19107, USA; eduardo.rame@jefferson.edu; 8Cardiovascular Outcomes, Quality, and Evaluative Research Center, University of Pennsylvania, Philadelphia, PA 19104, USA; 9American Heart Association, Dallas, TX 75231, USA; mariell.jessup@heart.org; 10The Lydia and Carol Kittner, Lea and Benjamin Davidai Division of Cardiovascular Medicine, Padeh-Poriya Medical Center, Azrieli Faculty of Medicine, Bar-Ilan University, Poriya 15208, Israel

**Keywords:** outcomes, left ventricular assist device

## Abstract

We aimed to identify a simple metric accounting for peri-procedural hemoglobin changes, independent of blood product transfusion strategies, and assess its correlation with outcomes in patients undergoing left ventricular assist device (LVAD) implantation We included consecutive patients undergoing LVAD implantation at a single center between 10/1/2008 and 6/1/2014. The anemia stress index (ASI), defined as the sum of number of packed red blood cells transfused and the hemoglobin changes after LVAD implantation, was calculated for each patient at 24 h, discharge, and 3 months after LVAD implantation. Our cohort included 166 patients (80.1% males, mean age 56.3 ± 15.6 years) followed up for a median of 12.3 months. Increases in ASI per unit were associated with a higher hazard for all-cause mortality and early RV failure. The associations between the ASI and all-cause mortality persisted after multivariable adjustment, irrespective of when it was calculated (adjusted HR of 1.11, 95% CI 1.03–1.20 per unit increase in ASI). Similarly, ASI at 24 h after implant was associated with early RV failure despite multivariable adjustment (OR 1.09, 95% CI 1.05–1.14). We present a novel metric, the ASI, that is correlated with an increased risk for all-cause mortality and early RV failure in LVAD recipients.

## 1. Introduction

Anemia is widely prevalent among patients with end-stage heart failure [1,2]. In patients undergoing left ventricular assist device (LVAD) implantation, anemia is further worsened by surgical blood loss, coagulopathy, device-related hemolysis, a generalized pro-inflammatory state, and the requirement for systemic anticoagulation [3]. Despite its high prevalence, the impact of anemia in LVAD patients remains poorly characterized with conflicting data on its association with morbidity and mortality [4,5,6].

As a consequence of peri-operative anemia at the time of LVAD implantation, blood transfusions are very common in this period. Although transfusion rapidly corrects anemia, studies assessing the impact of blood transfusions in patients undergoing cardiac surgery suggest a number of potential adverse effects, mediated both by volume and cytokines, resulting in the creation of a pro-inflammatory milieu, and the activation of leucocytes and coagulation factors, with an exacerbation of oxidative stress [7,8]. Accordingly, assessing the impact of anemia on outcomes in patients after LVAD implantation may be affected by transfusion strategies.

In this study, we assessed the impact of pre-implant anemia and changes in hemoglobin levels on outcomes of patients undergoing LVAD implantation, regardless of the transfusion strategy used. More specifically, we utilized a novel metric, the anemia stress index (ASI), defined as the sum of the number of packed red blood cells transfused and difference in hemoglobin levels pre- and post-LVAD implantation, and assessed its association with all-cause mortality and early right ventricular failure after LVAD implantation. The ASI assesses the actual change in hemoglobin after LVAD implantation and is not biased by transfusion strategies that frequently vary by physician. We hypothesized that the ASI calculated at different time points post-implant (at 24 h, on discharge, and at 3 months) would be predictive of all-cause mortality.

## 2. Methods

### 2.1. Study Population and Data Collection

This was a retrospective cohort study involving a single, large, tertiary care academic center. We included all consecutive patients who underwent continuous flow LVAD implantation at the Hospital of the University of Pennsylvania between 1 October 2008 and 1 June 2014 and survived for at least 24 h after the surgery, with data available on pre- and post-LVAD implant hemoglobin levels and the number of transfusions. All charts for inpatient hospitalization and outpatient visits were reviewed in detail by trained abstractors. Demographic, clinical, laboratory, peri-procedural, and imaging data were collected in detail via intensive chart review. The Institutional Review Board of the University of Pennsylvania approved the study protocol.

### 2.2. Independent Variable

We abstracted pre-implant hemoglobin, defined as the hemoglobin value available immediately prior to LVAD implantation. In addition, we collected the hemoglobin level at various time points after LVAD implantation—at 24 h, discharge, and at 3 months. Detailed data on the use of packed red blood cells (RBCs) during index hospitalization involving the LVAD implant were collected.

Using these collected variables, we calculated the anemia stress index (ASI; Figure 1), defined as:*Anemia stress index* = *pre-implant hemoglobin post-implant hemoglobin + number of packed RBCs transfused*

The rationale for utilizing the ASI is that it takes into account the actual change in hemoglobin occurring after LVAD implantation and is not biased by transfusion strategies, which frequently vary by physician. In other words, it allows for objective assessment of the change in hemoglobin following implantation which is not related to number of blood transfusions given during the implant (i.e., assessment of the amount of bleeding on one hand, and the ability to generate hemoglobin on the other hand). For example, if patient A had a pre-implant hemoglobin level of 12 g/dL, and after receiving 3 U of PRBC in the peri-operative period the post-implant hemoglobin at 3 months was 9 g/dL, the ASI value for this patient is 6. If patient B had a pre-implant hemoglobin of 8 g/dL, received 3 U PRBC in the peri-operative period, and had a post-implant hemoglobin of 9 g/dL, the anemia stress index value is 2. Although the amount of blood transfused for both these patients would be the same due to variable practice patterns, and the post-implant hemoglobin levels were the same as well, the ASI values are different and reflect the changes in hemoglobin levels in both patients.

### 2.3. Dependent Variable

Our primary outcome of interest was all-cause mortality. The secondary outcome of interest included early right ventricular failure after LVAD implantation, which was defined in accordance with the INTERMACS definition as the post-implantation use of inotropic therapy beyond 14 days or the need for right ventricular assist device (RVAD) implantation [9].

### 2.4. Covariates

Potential confounders for the association between the ASI and outcomes were included based on clinical rationale and prior studies [10]. Demographic covariates included age and sex. Clinical covariates included the pre-implant body mass index, diabetes, history of smoking, atrial fibrillation, chronic renal disease, chronic obstructive lung disease, and pulmonary hypertension. Implantation characteristics included prior cardiothoracic surgery and indication for LVAD implantation (classified as bridge to transplant, destination therapy, bridge to decision, or bridge to recovery).

### 2.5. Statistical Analysis

We stratified our study cohort into quartiles based on the distribution of the ASI at 24 h after LVAD implantation. Baseline characteristics were compared between these four groups using ANOVA for normally distributed continuous variables, Wilcoxon rank-sum test for non-normally distributed continuous variables, and chi-squared test for categorical variables.

To examine the association between the ASI at 24 h after LVAD implantation and all-cause mortality, we used a Cox proportional hazards model to estimate the hazard ratios (HRs) and their corresponding 95% confidence intervals (CIs) using the ASI as a continuous variable. The model was adjusted for the covariates listed above. We excluded patients with extreme ASI values (defined as those with an ASI value over the 90th percentile for the cohort), because extremes may bias generated estimates. The same analysis was repeated using the ASI at discharge and ASI at 3 months post-discharge. The proportional hazards assumption was evaluated and found to be met. In all analyses, participants were censored if the participant was lost to follow-up, underwent a cardiac transplant, had a device explant due to myocardial recovery, or reached the end of the follow-up period (1 December 2017).

To assess the association between the ASI at 24 h after implant and early right ventricular (RV) failure after LVAD implantation, a logistic regression model was fitted with the same covariates listed above.

All analyses were performed using STATA Statistical Software 11.2 (StataCorp, College Station, TX, USA). All reported *p* values are two sided, and a *p* value of <0.05 was considered statistically significant.

## 3. Results

Our study cohort comprised 166 patients who underwent LVAD implantation between 1 October 2008 and 1 June 2014. There were 133 (80.1%) males with a mean age of 56.3 ± 15.6 years. This included 78 (47%) patients with ischemic cardiomyopathy and 88 (53%) patients with non-ischemic, dilated cardiomyopathy. A HeartMate II device was implanted in 140 patients (84.3%), and the remaining 26 patients (15.7%) received a HeartWare LVAD. The implant strategy included 64 patients (38.6%) as bridge to transplant, 81 patients (48.8%) as destination therapy, 16 patients (9.6%) as bridge to decision, and 5 patients (3.0%) as bridge to recovery. Comorbidities were common, with diabetes prevalent in 45.2% and hypertension in 59.0% of the patients. The baseline median hemoglobin level was 11.1 (IQR9.7–12.3) g/dL. Baseline characteristics of the cohort stratified into quartiles by the ASI at 24 h after LVAD implantation are presented in Table 1.

### 3.1. Association between the ASI at 24 h after LVAD Implant and All-Cause Mortality

The median follow-up duration was 12.3 months (IQR 3.4—37.6); there were 86 deaths (51.8% patients) during this period. In the unadjusted analysis, we observed a statistically significant relationship between the ASI at 24 h post-LVAD implant and all-cause mortality (HR 1.05; 95% CI 1.03—1.07, *p* < 0.01; Figure 2). After multivariable adjustment, the association between the ASI and all-cause mortality over the follow-up period persisted, with an adjusted HR of 1.08 (95% CI 1.03—1.14, *p* < 0.01; Table 2). Other variables that correlated with time to mortality are included in Appendix A.

### 3.2. Association between the ASI at 24 h after LVAD Implant and Early Right Ventricular Failure

Early RV failure after LVAD implantation was seen in 32 (19.3%) patients. In the unadjusted model, there was a significant association between the ASI at 24 h post-implant and early RV failure post-LVAD implantation during the index hospitalization (OR 1.08; 95% CI 1.04–1.12; *p* < 0.01). Following multivariable adjustment, this association remained statistically significant (OR 1.09, 95% CI 1.04–1.14; *p* < 0.01; Table 3).

### 3.3. Association between the ASI at Discharge and All-Cause Mortality

At total of 139 (83.7%) patients survived the index hospitalization after LVAD implantation. In the unadjusted analysis, we observed a statistically significant relationship between the ASI at discharge and mortality (HR 1.04, 95% CI 1.01–1.07; *p* < 0.01). After multivariable adjustment, the association between the ASI and mortality over the follow-up period persisted, with an adjusted HR of 1.08 (95% CI 1.00–1.16, *p* = 0.04; Table 2).

### 3.4. Association between the ASI at 3 Months after LVAD Implant and All-Cause Mortality

A total of 128 (77.1%) patients survived longer than 3 months after LVAD implantation. In the unadjusted analysis, we observed a statistically significant relationship between the ASI at 3 months and all-cause mortality (HR 1.05, 95% CI 1.02–1.07; *p* < 0.01). After multivariable adjustment, the association between the ASI and mortality over follow-up persisted, with an adjusted HR of 1.09 (95% CI 1.02–1.16, *p* = 0.01; Table 2).

## 4. Discussion

We assessed the impact of a change in hemoglobin level regardless of the blood transfusion strategy on all-cause mortality and early post-implant right ventricular failure. We propose a novel and simple prognostic tool, called the “Anemia stress index”, which is a composite of the difference in hemoglobin prior to and after LVAD implantation and the number of packed red blood cells transfused. In our study, the ASI was correlated with all-cause mortality and early RV failure after LVAD implantation. This association persisted after multivariable adjustment for clinically significant confounders. Furthermore, the ASI was correlated with post-LVAD implant survival at all evaluated time points.

### 4.1. Prior Studies

Traditionally, anemia has been shown to be an independent marker of morbidity and all-cause mortality among patients with HF, with several studies showing a linear correlation between hemoglobin and mortality risk [2,11]. The pathogenesis of anemia in HF is multifactorial, and existing evidence suggests that it maybe secondary to renal dysfunction [12], with neurohormonal [13,14] and proinflammatory cytokine activation and bone marrow hypo-responsiveness [15]. The vast majority of HF patients also have some degree of iron deficiency [16]. Some studies suggest that LVADs may help to reverse the neurohormonal activation seen in HF; however, the prevalence of anemia among patients with LVADs remains high and has been reported to range from 46% to 80% in single-center cohorts [5,6,17,18].

Among factors contributing to the progression of anemia after LVAD implantation, peri-operative bleeding plays a major role, in addition to ongoing inflammation, hemolysis, and gastrointestinal bleeding [5]. Although blood transfusions provide a rapid, short-term solution to anemia, observational studies suggest an association between transfusions and increased rates of infection, ischemic complications, RV failure, and death in patients undergoing coronary artery bypass grafting [7,8]. Our previous study showed a positive correlation between blood transfusions and one-year mortality in LVAD patients [6,19,20]. Furthermore, practice patterns with regard to transfusion thresholds vary significantly across providers. Accordingly, assessing the degree of anemia in patients with LVADs must include an estimation of the amount of blood products received.

### 4.2. Study Implications

Our study highlights the importance of calculating ASI values for every LVAD patient as an additional prognostic factor, irrespective of the different transfusion strategies employed. It is a simple and easy tool which can be readily used to risk-stratify patients after LVAD implantation. It correlated with all-cause mortality irrespective, of the time after LVAD implantation when it was calculated. Future randomized studies are needed to evaluate strategies of decreasing the ASI score by increasing the post-implant hemoglobin levels, such as by treating LVAD candidates with intravenous iron therapy. Our novel ASI score enables estimation of the blood loss and hemoglobin generation, regardless of the transfusion strategy.

### 4.3. Study Limitations

The results of our study should be interpreted in the light of several potential limitations. First, we did not have information regarding the etiology of anemia. However, our study has shown that ASI is an effective prognostic tool and reflects real-world practice and outcomes. Second, our study was observational in nature and we cannot establish causality, and the possibility of residual confounding exists. For example, we do not have more granular data on peri-operative details such as cross clamp time; nonetheless, we utilized a large number of covariates that influence outcomes after LVAD implantation, and this has likely reduced this source of error. Third, we do not have longer-term follow-up data available on our cohort to provide data on longer-term outcomes.

## 5. Conclusions

In conclusion, assessing the impact of anemia on the outcomes of LVAD patients is limited by blood product transfusion strategies which are frequently not standardized. We present a novel metric comprising the sum of the difference in hemoglobin levels prior to and after LVAD implantation with amount of packed RBC transfused (anemia stress index), which correlated with both all-cause mortality and early right ventricular failure after LVAD implantation. Accordingly, using the anemia stress index can help to risk-stratify patients after LVAD implantation. Additional studies are needed to explore therapeutic strategies which will result in a lower anemia stress index in an effort to improve outcomes for patients on durable LVAD support.

## Figures and Tables

**Figure 1 jcm-11-04517-f001:**
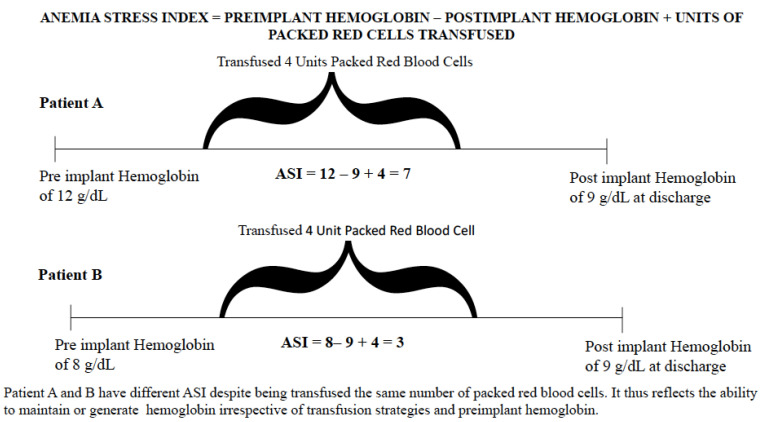
Definition of the anemia stress index.

**Figure 2 jcm-11-04517-f002:**
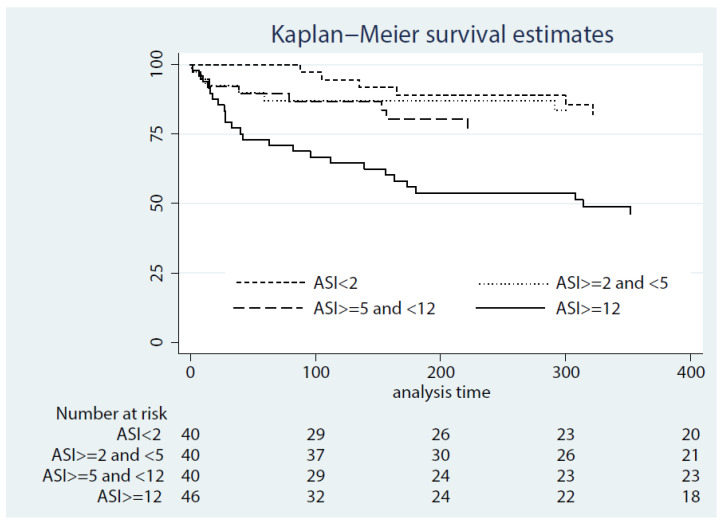
Kaplan–Meier survival estimates of the cohort stratified by quartiles of the anemia stress index at 24 h post-LVAD implantation.

**Table 1 jcm-11-04517-t001:** Baseline characteristics of the cohort stratified by the ASI at 24 h after LVAD implantation.

Characteristic	Quartile 1 ASI < 2 (*n* = 40)	Quartile 2 ASI > 2 to <5 (*n* = 40)	Quartile 3 ASI > 5 to <12 (*n* = 40)	Quartile 4 ASI > 12 (*n* = 46)	*p* Value
Age, years	51.7 ± 14.9	52.5 ± 17.9	61.1 ± 15.4	59.6 ± 12.6	<0.01
Male sex	28 (70%)	30 (75%)	32 (80%)	43 (93.5%)	0.04
Race					0.26
Caucasian	17 (42.5%)	16 (40%)	15 (37.5%)	29 (63%)	
African-American	6 (15.1%)	10 (25%)	6 (15%)	9 (19.6%)	
Other	3 (7.5)	1 (2.5%)	1 (2.5%)	0 (0)	
Unknown	14 (35%)	13 (32.5%)	18 (45%)	8 (17.4%)	
BMI (mean ± SD)	30.5 ± 7.4	27.3 ± 6.1	27.7 ± 7.3	28.6 ± 5.4	0.13
Diabetes	19 (47.5%)	16 (40%)	18 (45%)	22 (47.8%)	0.88
Hypertension	25 (62.5%)	15 (37.5%)	23 (57.5%)	35 (76.1%)	<0.01
Dyslipidemia	26 (65%)	25 (62.5%)	27 (67.5%)	30 (65.2%)	0.97
Atrial fibrillation	13 (32.5%)	14 (35%)	18 (45%)	23 (50%)	0.31
Stroke/TIA	8 (20%)	3 (7.5%)	4 (10%)	8 (17.4%)	0.31
Chronic renal disease	10 (25%)	15 (37.5%)	13 (32.5%)	20 (43.5%)	0.30
Prior CT surgery	10 (25%)	10 (25.0%)	18 (45%)	18 (39.1%)	0.13
HF etiology					0.60
Ischemic cardiomyopathy	17 (42.5%)	16 (40%)	19 (47.5%)	26 (56.5%)	
Non ischemic cardiomyopathy:					
-Idiopathic	18	19	16	17	
-Congenital heart disease	0	1	0	0	
-Viral cardiomyopathy	2	0	0	0	
-Peripartum cardiomyopathy	1	2	1	0	
-Alcohol-induced cardiomyopathy	0	1	0	0	
-Myocarditis	0	0	1	1	
-Chemotherapy-induced cardiomyopathy	1	1	2	1	
-Valvular cardiomyopathy	1	0	1	1	
LVAD reason					0.93
Cardiogenic shock	21 (52.5%)	17 (43.6%)	19 (47.5%)	22 (50%)	
Inotrope-dependent heart failure	14 (35%)	14 (35.9%)	15 (37.5%)	14 (31.8%)	
Worsening, non-inotrope-dependent heart failure	4 (10%)	7 (17.9%)	5 (12.5%)	7 (15.9%)	
Intractable ventricular arrhythmia	0 (0)	1 (2.6%)	1 (2.5%)	1 (2.3%)	
Pump exchange	1 (2.5%)	0 (0)	0 (0)	0 (0)	
LVAD indication					0.01
Bridge to transplant	23 (57.5%)	13 (32.5%)	11 (27.5%)	17 (36.9%)	
Bridge to decision	6 (15%)	2 (5%)	4 (10%)	4 (8.7)	
Destination therapy	11 (27.5%)	21 (52.5%)	24 (60%)	25 (54.3%)	
Bridge to recovery	0 (0)	4 (10%)	1 (2.5%)	0 (0)	
LVAD type					0.28
HeartMate II	30 (75%)	36 (90%)	34 (85%)	40 (86.9%)	
HeartWare	10 (25%)	4 (10%)	6 (15%)	6 (13%)	
Pulmonary hypertension	13 (32.5%)	16 (40%)	16 (40%)	11 (24.4%)	0.37
COPD	9 (23.1%)	3 (7.5%)	8 (20%)	9 (19.6%)	0.27
History of smoking	16 (40%)	17 (42.5%)	16 (40%)	19 (41.3%)	0.99
Hemoglobin baseline (median, IQR)	11.2 (8.9–12.3)	11.4 (10.0–12.9)	11.1 (10.2–12.2)	10.6 (9.4–12.0)	0.18
Platelet baseline (mean ± SD)	194.6 ± 79.6	178.3 ± 58.2	191.3 ± 64.2	192.4 ±75.1	0.72
Albumin (mean ± SD)	3.4 ± 0.6	3.4 ± 0.5	3.4 ± 0.5	3.3 ± 0.7	0.71

BMI—body mass index; COPD—chronic obstructive pulmonary disease; CT—cardiothoracic; IQR—interquartile range; LVAD—left ventricular assist device; SD—standard deviation; TIA—transient ischemic attack.

**Table 2 jcm-11-04517-t002:** Results of the Cox proportional hazards model exploring the association between the ASI and all-cause mortality.

	Unadjusted HR (95% CI) for Mortality	Adjusted HR (95% CI) for Mortality
Anemia stress index at 24 h post-implant	1.09 (1.04–1.14)	1.08 (1.03–1.14)
Anemia stress index at discharge	1.04 (1.01–1.07)	1.08 (1.00–1.16)
Anemia stress index at 3 months after LVAD implant	1.05 (1.02–1.07)	1.09 (1.02–1.16)

**Table 3 jcm-11-04517-t003:** Results of the multivariable model exploring the association between the ASI at 24 h post-LVAD implant and early right ventricular failure after LVAD implantation.

Characteristic	OR (95% CI)	*p* Value
Anemia stress index * at 24 h	1.09 (1.04–1.14)	<0.01
Age *	0.99 (0.95–1.03)	0.53
Male sex	1.25 (0.37–4.24)	0.72
BMI *	0.99 (0.91–1.06)	0.73
Diabetes	1.07 (0.42–2.74)	0.89
COPD	2.30 (0.74–7.21)	0.74
Pulmonary hypertension	1.06 (0.40–2.80)	0.40
Smoking history	0.62 (0.24–1.64)	0.34
Atrial fibrillation	1.35 (0.51–3.57)	0.55
Chronic renal disease	1.50 (0.55–4.10)	0.55
Prior CT surgery	1.01 (0.39–2.64)	0.98
LVAD indication **		
Bridge to transplant	Reference	
Bridge to decision	1.06 (0.22–5.16)	0.94
Destination therapy	0.76 (0.26–2.23)	0.62

BMI—body mass index; COPD—chronic obstructive pulmonary disease; CT—cardiothoracic; LVAD—left ventricular assist device. * Treated as a continuous variable; ** Stratified into bridge to transplant, bridge to recovery, bridge to decision, and destination therapy.

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
