# Peer review of "The Anemia Stress Index—Anemia, Transfusions, and Mortality in Patients with Continuous Flow Ventricular Assist Devices"

_jcm, 2022, doi:10.3390/jcm11154517_

Round 1
Reviewer 1 Report
Authors aimed to analyze the prognostic value of anemia stress index (ASI) in subject after the LVAD implantation procedure. this is a novel factor which may determine the burden of blood transfusion. This is an interesting work which may be produce a valuable clinical implications in patients who undergo the surgeries. However, it need to clarified in few points:
1. What was the etiology of heart failure among study cohort despite ischemic (DCM, HCM,RCM)?
2. Were there any differences in clinical presentation, mortality and ASI between type of LVAD implanted, indication for LVAD or duration of the disease? Different operation requirements may cause higher blood loss and higher ASI. Also, anemia in long-lasting heart failure is known and may also produce higher blood requirement after the surgery.
3. What was the prognostic role of ASI in longer observation periods (2,3years or longer). This work was based on population who had been operated between more than decade ago. Hence, the data should be available by now.
4. Authors include large number of indices in their comparisons; shouldn’t you use the Bonferroni correction in significance calculations?
5. Inclusion of Kaplan-Meyer analysis (between the ASI quartiles) may be very interesting for a potential reader.
6. The discussion is rather short. Which generally is good; however, authors should explain their own findings in the light of previous works rather than describing prior studies only.
Author Response
Authors aimed to analyze the prognostic value of anemia stress index (ASI) in subject after the LVAD implantation procedure. this is a novel factor which may determine the burden of blood transfusion. This is an interesting work which may be produce a valuable clinical implications in patients who undergo the surgeries. However, it need to clarified in few points:
We would like to thank the reviewer for their comments. Please find an itemized response to all comments below:
- What was the etiology of heart failure among study cohort despite ischemic (DCM, HCM,RCM)?
Etiology for all patients with non-ischemic cardiomyopathy was dilated cardiomyopathy. We now provide additional details on cause for the non ischemic cardiomyopathy in Table 1 as well. We have added this to our manuscript which now states:
“This included 78 (47%) patients with ischemic cardiomyopathy and 88 (53%) patients with non-ischemic, dilated cardiomyopathy.”
- Were there any differences in clinical presentation, mortality and ASI between type of LVAD implanted, indication for LVAD or duration of the disease?
As displayed in Table 1, we did not note any difference between type of LVAD or the reason for LVAD implantation across different ASI quartiles. However, patients in the lowest ASI quartile were more likely to receive their LVAD as a bridge to transplant strategy.
We have also added data from our Cox proportional hazards model showing association between various covariates included in our model examining association between mortality and ASI. This includes all variables suggested by the reviewer. In our model, variables significantly associated with mortality included ASI, pre-implant body mass index, pre-implant chronic kidney disease and type of LVAD with a better survival associated with HeartMate 2 as opposed to HeartWare. These results are provided in supplemental appendix.
|
Variable |
HR (95% CI) |
|
ASI at 24 hours |
1.10 (1.04-1.16) |
|
Age at Implant, per year increase |
1.03 (1.00-1.05) |
|
Male Sex |
0.78 (0.37-1.63) |
|
Pre-Implant Body Mass Index, per unit increase |
1.07 (1.02-1.12) |
|
Diabetes Mellitus |
0.62 (0.35-1.09) |
|
Chronic Obstructive Pulmonary Disease |
1.04 (0.57-1.97) |
|
History of smoking |
1.50 (0.85-2.67) |
|
Atrial fibrillation |
0.77 (0.44-1.34) |
|
Pre-Implant Chronic Kidney Disease |
2.21 (1.27-3.85) |
|
Previous sternotomy |
0.63 (0.36-1.10) |
|
Indication for LVAD Implantation Bridge to Transplant Destination Therapy Bridge to Decision |
Reference 2.44 (0.93-6.42) 0.70 (0.64-2.53) -* |
|
Type of LVAD implanted HM2 HM3 |
Reference 3.03 (1.26-7.29) |
- Different operation requirements may cause higher blood loss and higher ASI. Also, anemia in long-lasting heart failure is known and may also produce higher blood require ement after the surgery.
We agree with the reviewer and have acknowledged these factors in our discussion section.
- What was the prognostic role of ASI in longer observation periods (2,3years or longer). This work was based on population who had been operated between more than decade ago. Hence, the data should be available by now.
We unfortunately no longer have access to clinical data on this cohort and acknowledge this as a limitation. Our limitations section now states:
“Third, we do not have longer term follow-up data available on our cohort to provide data on longer term outcomes.”
- Authors include large number of indices in their comparisons; shouldn’t you use the Bonferroni correction in significance calculations?
We used a total of 12 covariates in our regression model and have a total number of 166 patients in our cohort. As one covariate per ten patients is acceptable, we do not believe that our model is overfitted and hence no additional correction is required.
- Inclusion of Kaplan-Meyer analysis (between the ASI quartiles) may be very interesting for a potential reader.
We thank the reviewer for this comment and have now provided KM curves for ASI quartiles.
- The discussion is rather short. Which generally is good; however, authors should explain their own findings in the light of previous works rather than describing prior studies only.
We thank the reviewer for this comment. We aimed to discuss our main findings in the clinical context and made some edits in the manuscript according to the reviewer suggestion.
Reviewer 2 Report
The article presents a retroprospective single center study of 166 patients who underwent LVAD implantation from 2008 to 2014. The authors tried to investigate the correlation between death and the ASI index (anemia stress index). Pzrimary and secondary outcomes were survival and right ventricle disfunction. Congratulations to the authors for their clinical results and surgical outcomes.
Some comments:
Comment 1:
The authors should add KM curves for survival and freedom from RV failure.
Comment2: e
The authors should add more surgical details (ECC and xclamp time; cardioplegia; conversion; reclamp) and describe the postoperative course con complications
.
Comment 3:
Probably the authors can try to add a ROC analysis to find out a ASI value to discriminate high and low risk patients.
Comment 4:
What about echocardiography FU, postoperative results were stable during the 6 postoperative months?
Comment 5:
Can the authors extend their FU?
Author Response
Comment 1:
The authors should add KM curves for survival and freedom from RV failure.
We thank the reviewer for this suggestion. We have now added KM curves for survival. We are unable to provide KM curves for RV failure as this was assessed within the first 14 days of LVAD implantation.
Comment2:
The authors should add more surgical details (ECC and xclamp time; cardioplegia; conversion; reclamp) and describe the postoperative course con complications
We do not have these details available in our database and acknowledge this as a limitation of our study.
Comment 3:
Probably the authors can try to add a ROC analysis to find out a ASI value to discriminate high and low risk patients.
The primary purpose of our analysis was to describe anemia stress index as a risk predictor for mortality and early right ventricular failure in LVAD recipients. Accordingly, odds ratio and hazards ratio described predict the probability of these events. ROC curve that is based on sensitivity and specificity would not be appropriate for this analysis.
Comment 4:
What about echocardiography FU, postoperative results were stable during the 6 postoperative months?
We do not have these details in our database and are unable to provide them.
Comment 5:
Can the authors extend their FU?
We are unable to extend our follow-up further and acknowledge this as a limitation that states:
“Third, we do not have longer term follow-up data available on our cohort to provide data on longer term outcomes.”